# Parallel Spiking Neurons with High Efficiency and Ability to Learn Long-term Dependencies

**Wei Fang**[1,2], **Zhaofei Yu**[4], **Zhaokun Zhou**[3,2], **Ding Chen**[5,2], **Yanqi Chen**[1,2],
**Zhengyu Ma**[2*], **Timothée Masquelier**[6], **Yonghong Tian**[1,2,3*]

[1]School of Computer Science, Peking University, China
[2]Peng Cheng Laboratory, China
[3]School of Electronic and Computer Engineering, Shenzhen Graduate School, Peking University, China
[4]School of Artificial Intelligence, Peking University, China
[5]Department of Computer Science and Engineering, Shanghai Jiao Tong University, China
[6]Centre de Recherche Cerveau et Cognition (CERCO), UMR5549 CNRS - Univ. Toulouse 3, France

## Abstract

Vanilla spiking neurons in Spiking Neural Networks (SNNs) use charge-fire-reset neuronal dynamics, which can only be simulated serially and can hardly learn long-time dependencies. We find that when removing reset, the neuronal dynamics can be reformulated in a non-iterative form and parallelized. By rewriting neuronal dynamics without reset to a general formulation, we propose the Parallel Spiking Neuron (PSN), which generates hidden states that are independent of their predecessors, resulting in parallelizable neuronal dynamics and extremely high simulation speed. The weights of inputs in the PSN are fully connected, which maximizes the utilization of temporal information. To avoid the use of future inputs for step-by-step inference, the weights of the PSN can be masked, resulting in the masked PSN. By sharing weights across time-steps based on the masked PSN, the sliding PSN is proposed to handle sequences of varying lengths. We evaluate the PSN family on simulation speed and temporal/static data classification, and the results show the overwhelming advantage of the PSN family in efficiency and accuracy. To the best of our knowledge, this is the first study about parallelizing spiking neurons and can be a cornerstone for the spiking deep learning research. Our codes are available at `https://github.com/fangwei123456/Parallel-Spiking-Neuron`.

## 1 Introduction

Spiking Neural Networks (SNNs) are the next generation [1] of Artificial Neural Networks (ANNs) using a lower abstraction of the biological neural system. The spiking neurons are the key components of SNNs, which process input currents with complex neuronal dynamics and fire spikes as outputs when their membrane potential reaches the threshold. The SNNs use discrete spikes to communicate between layers, which enable the event-driven computational paradigm and have extremely high power efficiency in neuromorphic chips such as True North[2], Loihi [3] and Tianjic [4]. Due to their high biological plausibility, SNNs have been regarded by neuroscientists as useful tools for analyzing, simulating, and learning the biological system [5, 6]. With the introduction of deep learning methods [7, 8, 9, 10, 11, 12], the performance of SNNs on real-world tasks has been greatly improved and the applications of SNNs are expanded [13, 14]. As the bridge between neuroscience and computational science, SNNs have attracted more and more research interest in recent years.

---

*Corresponding author

37th Conference on Neural Information Processing Systems (NeurIPS 2023).

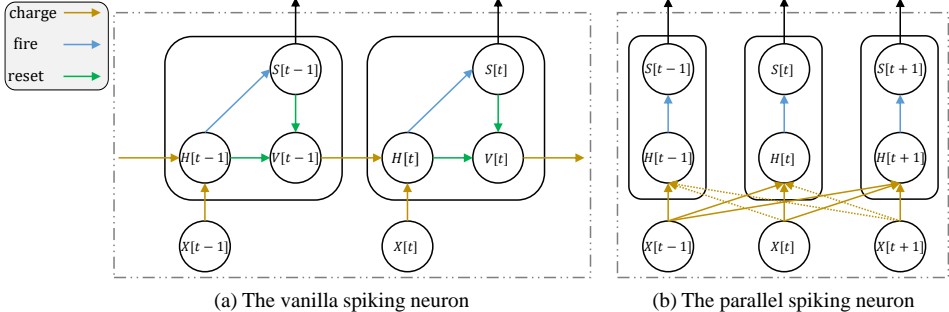

(a) The vanilla spiking neuron          (b) The parallel spiking neuron

Figure 1: The computational graphs of the vanilla spiking neuron and the parallel spiking neuron. Figure (a) is cited from [15]. The dotted lines in Figure (b) are weights that can be masked for step-by-step computation. $X[t], S[t]$ are the input current and the output spike, and $H[t], V[t]$ are hidden states at time-step $t$.

The binary characteristic of spikes causes a lot of information losses, which is the main factor leading to the lower accuracy of SNNs than ANNs. Among efforts to bridge the performance gap, the improvement of spiking neurons is a universal approach and has been widely explored [15, 16, 17, 18, 19, 20, 21]. However, previous works are limited to the existing serial charge-fire-reset computing paradigm shown in Fig.1(a), which suffers from slow simulation speed and can hardly learn long-term dependencies.

In this paper, we propose the Parallel Spiking Neuron (PSN) and its variants, the masked PSN as well as the sliding PSN (SPSN). Fig.1(b) shows the computational graph of the PSN, which uses a non-predecessor dependency method to generate hidden states, leading to parallelizable neuronal dynamics and extremely high simulation speed. Since the connections from inputs to $H[t]$ are direct, the PSN has a better learning ability for long-term dependencies than other vanilla spiking neurons. To generate $H[t]$ immediately when $X[t]$ arrives, connections from $X[i], i \geq t+1$ can be masked, as the dotted line shown in Fig.1(b), and then the neuron becomes the masked PSN. The parameters of the masked PSN can also be set as time-invariant, which derives the sliding PSN and is more flexible for input sequences with variable lengths. The main contributions of this paper are as follows:

1) We analyze the influence of removing reset from the widely used charge-fire-reset neuronal dynamics, showing that the spiking neuron can be parallelized without reset.

2) By rewriting the neuronal dynamics without reset to a general formula, we obtain the PSN with fully parallelizable neuronal dynamics. For step-by-step serial forward and variable length sequences processing, the masked PSN and the sliding PSN are also derived.

3) We compare the simulation speed of the PSN family and vanilla spiking neurons, showing an overwhelming performance advantage of the parallel neuronal dynamics. We evaluate the PSN family on sequential, static, and neuromorphic data classification tasks and achieve higher accuracy than previous spiking neurons.

## 2   Related Work

### 2.1   Deep Learning for Spiking Neural Networks

The ANN-to-SNN conversion (ANN2SNN) [22, 23, 24, 25, 26] and the surrogate training [27, 28] are the two main deep learning methods for SNNs. The ANN2SNN method uses firing rates in SNNs to approximate the activations in ANNs by converting weights from ANNs to SNNs with extra normalizations to reduce errors. It achieves close performance to that of ANNs in the ImageNet dataset [29], but requires many time-steps to estimate accurate firing rates. The surrogate training method re-defines the gradient of the Heaviside function used in the spike generating process by a smooth and differentiable surrogate function. With the surrogate gradients, the SNNs can be trained by backpropagation through time (BPTT) and gradient descent. The surrogate training method is not based on rate coding as ANN2SNN, resulting in fewer time-steps. However, the use of BPTT during training requires more training time and memory usage than ANN2SNN.

## 2.2 Improvement of Spiking Neurons

Improving the spiking neurons inspired by biological mechanisms or techniques from ANNs is a practical method. The Parametric Leaky Integrated-and-Fire (PLIF) spiking neuron [15] incorporates learnable membrane time constants and is able to combine the learning of both synapse weights and neuronal dynamics. The gated LIF (GLIF) spiking neuron [16] adds learnable gates to fuse different integration, decay, and reset methods, which enlarges the representation space and increases the heterogeneity as well as adaptivity of spiking neurons. The $k$-based Leaky Integrate-and-Fire (KLIF) neuron [18] adds a learnable scaling factor to adjust the surrogate gradient and applies a ReLU activation to avoid the negative membrane potential. The Multi-Level Firing (MLF) unit [19] contains LIF neurons with thresholds in different levels, which relieves the gradient vanishing problem and improves expression ability. Ponghiran et al. [20] improve the spiking neurons by adding input gates, forget gates, recurrent connections, and multi-bit outputs, which reduce the mismatch between actual and surrogate gradients.

## 2.3 Improvement of Training Methods and Network Structure

Beyond the spiking neurons, modifying training techniques and network structure are also effective methods to promote the performance of SNNs.

**Normalization** The threshold-dependent batch normalization (TDBN) [30] normalizes features in both temporal and spatial dimensions and eliminates the variance introduced by thresholds, which is a more suitable normalization method than the batch normalization [31] for SNNs. Batch Normalization Through Time (BNTT) [32] decouples BN along time-steps and uses time-wise parameters, which estimates the distribution of temporal-variant inputs more precisely. Temporal Effective Batch Normalization (TEBN) [33] adds learnable affine transformation on each time-step and has a better ability to capture temporal distributions.

**Training Techniques** Deep Continuous Local Learning (DECOLLE) [34] uses local losses to avoid BPTT and implements online learning for SNNs. Online training through time (OTTT) [35] approximates iterative gradients with temporal dependencies by eligibility traces updating with time-steps, which also achieves online training and only requires constant training memory agnostic to time-steps. Sparse spiking gradient descent [36] only uses neurons whose membrane potentials are inside a given range for backpropagation, which achieves a speedup and memory reduction in the backward. The attention mechanism on membrane potentials [37] eliminates redundant information, decreases firing rates, and reduces overfitting. Dspike [38] uses the finite difference to estimate gradients and achieves higher accuracy than plain surrogate gradient methods. The temporal efficient training (TET) [39] calculates loss at each time-step and averages them rather than using averaged outputs to calculate loss, which improves the temporal scalability of SNNs.

**Network Structure** Spike-Element-Wise ResNet [40] solves the gradient vanishing/exploding problems of the plain Spiking ResNet caused by sigmoid-like surrogate functions, which successfully trained the first deep SNN with more than 150 layers. Spikformer [41] modifies the softmax-based self-attention in Transformer [42] to the spike-based formulation, which is appropriate for SNNs and achieves state-of-the-art accuracy on the ImageNet dataset.

## 2.4 Acceleration for Sequence Processing

Massively parallel computing with graphics processing units (GPUs) [43] is one of the key factors that drives deep learning research. However, the serial computing characteristic of the recurrent structure such as SNNs and Recurrent Neural Networks (RNNs) can not fully exploit the parallel computing power of GPUs. To accelerate the sequence processing, convolution-based methods [44, 45] discard the recurrent structure and employ the convolutional layers, which are fully parallelized during training on GPUs. The gated impulse linear recurrent network [46] is another solution that uses the Parallel Prefix Sum (Scan) algorithm [47] to parallelize linear recurrences. The Transformer [42] replaces recurrences with self-attention and positional encoding, which achieves faster training speed than recurrent and convolutional encoder-decoder structures.

# 3 Methods

In this section, we introduce the motivation of removing neuronal reset, the idea of generalizing spiking neurons without reset as the PSN, and the derivation of the masked PSN and the sliding PSN. Note that we use regular letters such as $X$ to represent scalars, and bold letters such as $\boldsymbol{X}$ to represent tensors.

## 3.1 Vanilla Spiking Neurons W/WO Reset

Spiking neurons in SNNs have rich neuronal dynamics, which endow SNNs with temporal information processing ability. In most cases, the behaviors of spiking neurons can be described by three discrete-time equations [15]:

$$H[t] = f(V[t-1], X[t]), \tag{1}$$
$$S[t] = \Theta(H[t] - V_{th}), \tag{2}$$
$$V[t] = \begin{cases} H[t] \cdot (1 - S[t]) + V_{reset} \cdot S[t], & \text{hard reset} \\ H[t] - V_{th} \cdot S[t], & \text{soft reset} \end{cases}, \tag{3}$$

where $X[t]$ is the input current, $H[t]$ is the membrane potential after charging but before firing, $V[t]$ is the membrane potential after firing, and $S[t]$ is the output spike at time-step $t$. $V_{th}$ in Eq.(2) is the threshold potential, and $V_{reset}$ in Eq.(3) is the reset potential. $\Theta(x)$ is the Heaviside step function and $\Theta(x) = 1$ for all $x \geq 0$, otherwise $\Theta(x) = 0$. Eq.(1) is the neuronal charging equation, and $f$ is specific to different spiking neurons. After charging, $H[t]$ will be compared to $V_{th}$ and determine whether to fire spikes, which is described by Eq.(2). After firing, the membrane potential will be reset, as Eq.(3) shows. Note that there are two kinds of resets, which are the hard reset and the soft reset. If the neuron fires, the hard reset will set $V[t]$ to $V_{reset}$, while the soft reset will decrease $V[t]$ by $V_{th}$. The hard reset is widely used in surrogate training for better performance observed in experiments [48], while the soft reset is preferred in ANN2SNN for lower conversion errors. When simulating SNNs, the iterative process following Eqs.(1)-(3) over time-steps is employed, which has a time complexity of $\mathcal{O}(T)$, where $T$ is the number of time-steps.

For frequently-used spiking neurons, Eq.(1) is linear and can be reformulated to a non-iterative equation if we can ignore Eq.(3) in some cases, e.g., in the subthreshold regime $H[t] < V_{th}$ for all $t$ during a period. Then $V[t] = H[t]$ at all $t$ and we will only use $H[t]$ in this case. More specifically, we take the Integrate-and-Fire (IF) neuron and the Leaky Integrate-and-Fire (LIF) neuron as examples. For simplicity, suppose $H[-1] = 0$ for both types of neurons, and the resting potential for the LIF neuron is 0. The neuronal charge equation, or Eq.(1), for the IF neuron is

$$H[t] = H[t-1] + X[t]. \tag{4}$$

And Eq.(4) can be easily reformulated as

$$H[t] = \sum_{i=0}^{t} X[i]. \tag{5}$$

The neuronal charge equation for the LIF neuron is

$$H[t] = (1 - \frac{1}{\tau_m}) \cdot H[t-1] + \frac{1}{\tau_m} \cdot X[t], \tag{6}$$

where $\tau_m$ is the membrane time constant. Eq.(6) can also be reformulated as

$$H[t] = \frac{1}{\tau_m} \cdot \sum_{i=0}^{t} (1 - \frac{1}{\tau_m})^{t-i} \cdot X[i]. \tag{7}$$

With the non-iterative equations, $H[t]$ at all time-steps, or $\boldsymbol{H} = \{H[0], H[1], ..., H[T-1]\}$, can be calculated in parallel. For each $H[t]$ obtained by the cumulative sum operation, the time complexity can be as low as $\mathcal{O}(\log(t)) \leq \mathcal{O}(\log(T))$ when using the Parallel Prefix Sum (Scan) algorithm. When $\boldsymbol{H}$ is given, the element-wise operation Eq.(2) can also be applied on the whole $\boldsymbol{H}$ and the

time complexity is $\mathcal{O}(1)$ in devices that support parallel computing such as the CUDA devices. Thus, the whole time complexity for the neuronal charging and firing reduces to $\mathcal{O}(\log(T))$ when we can ignore the neuronal resetting.

So, how can we ignore neuronal resetting at all time-steps? A possible method is setting $V_{th} = +\infty$ to make Eq.(3) become an identity function. However, the neuron can only output 0 in this case, which is meaningless. Another method is to remove the neuronal resetting directly from the neuronal dynamics, which means only using Eq.(1) and Eq.2. This crude and simple method may raise further concerns that the absence of neuronal resetting will cause the neuron to fire uninterruptedly. Note that the firing rate is determined by inputs, charging, thresholds, and resetting. Thus, the effect of only removing reset on firing rates is not deterministic. In addition, the experiment results in the next section will show that the uninterrupted firing issue will not occur.

## 3.2 Parallel Spiking Neuron

After removing neuronal resetting, the charging equation of the spiking neuron can be formulated into a non-iterative equation, and the solution for $H[t]$ becomes a cumulative sum operation. It can be found that Eq.(5) and Eq.(7) are two specific cases of a linear combination of $X[i]$. More specifically, we formulate a general linear combination as

$$H[t] = \sum_{i=0}^{T-1} W_{t,i} \cdot X[i], \tag{8}$$

where $W_{t,i}$ is the weight between $X[i]$ and $H[t]$. Then, $W_{t,i} = \Theta(t-i)$ for the IF neuron without reset, and $W_{t,i} = \frac{1}{\tau_m}(1 - \frac{1}{\tau_m})^{t-i} \cdot \Theta(t-i)$ for the LIF neuron without reset. Based on the above analysis, we propose the Parallel Spiking Neuron (PSN), whose neuronal dynamics are as follows:

$$\boldsymbol{H} = \boldsymbol{W}\boldsymbol{X}, \qquad \boldsymbol{W} \in \mathbb{R}^{T \times T}, \boldsymbol{X} \in \mathbb{R}^{T \times N} \tag{9}$$

$$\boldsymbol{S} = \Theta(\boldsymbol{H} - \boldsymbol{B}), \quad \boldsymbol{B} \in \mathbb{R}^{T}, \boldsymbol{S} \in \{0,1\}^{T \times N} \tag{10}$$

where $\boldsymbol{X}$ is the input sequence, $\boldsymbol{W}$ is the learnable weight matrix, $\boldsymbol{H}$ is the hidden state sequence, $\boldsymbol{B}$ is the learnable threshold, and $\boldsymbol{S}$ is the binary output spike sequence. $N$ is the number of batch size, and $T$ is the number of time-steps. Note that $\boldsymbol{W}$ is a $T \times T$ matrix, indicating that $H[t]$ can integrate the information from all time-steps directly, rather than that of the last one or two time-steps. Thus, the PSN is a $T$-order neuron. No iterative equation is used, and the computation of PSN is fully parallel. The time complexity of the PSN is determined by Eq.(9), which is a matrix-matrix multiplication. The simulation of PSN is much faster than that of vanilla spiking neurons because the matrix-matrix multiplication is highly optimized in linear algebra libraries such as Intel MKL and cuBLAS. BPTT is still used during training, but also in a parallel, rather than a serial formulation.

## 3.3 $k$-Order Masked Parallel Spiking Neuron

The high-order characteristic of the PSN is also a double-edged sword. The output sequence $\boldsymbol{S}$ can only be generated when all $X[t]$ have arrived, indicating that each spiking neuron layer will increase latency $T$ to the whole SNN. Such a latency problem is also reported on some time-to-first-spike SNNs [49, 50, 51]. Note that although the latency is unavoidable, the throughput can still approach the plain SNN with the multi-stage pipeline [49]. To solve the latency problem of the PSN, we add a mask $\boldsymbol{M}_k$ to multiply $\boldsymbol{W}$ element-wisely before generating $\boldsymbol{H}$ and propose the $k$-order masked PSN, whose neuronal charging equation is

$$\boldsymbol{H} = (\boldsymbol{W} \cdot \boldsymbol{M}_k)\boldsymbol{X}, \qquad \boldsymbol{W} \in \mathbb{R}^{T \times T}, \boldsymbol{M}_k \in \mathbb{R}^{T \times T}, \boldsymbol{X} \in \mathbb{R}^{T \times N} \tag{11}$$

where $\boldsymbol{M}_k$ is defined as

$$\boldsymbol{M}_k[i][j] = \begin{cases} 1, & j \le i \le j + k - 1 \\ 0, & \text{otherwise} \end{cases}. \tag{12}$$

With the mask $\boldsymbol{M}_k$, $H[t]$ depends only on the latest $k$ inputs $\{X[t-k+1], ..., X[t-1], X[t]\}$ (we assume that $X[i] = 0$ for all $i < 0$) and $S[t]$ can be computed and sent to the next layer once $X[t]$ is received.

When training the $k$-order masked PSN, the progressive masking method is employed, which uses an all-ones matrix $\mathbf{1}$ as the mask first, and gradually replaces $\mathbf{1}$ with $\mathbf{M}_k$. More specifically, we use

$$\mathbf{M}_k(\lambda) = \lambda \cdot \mathbf{M}_k + (1 - \lambda) \cdot \mathbf{1} \tag{13}$$

as the mask, and $\lambda$ increases from 0 to 1 during training. In the early stages of training, $\mathbf{M}_k(\lambda) \approx \mathbf{1}$, indicating that the future information can be exploited to provide appropriate primary parameters and help the network converge.

### 3.4 $k$-Order Sliding Parallel Spiking Neuron

The parameters of the PSN and the masked PSN are time-wise, which require extra operations for processing sequences with variable lengths. To solve this issue, we modify the parameters to be shared across time-steps and propose the $k$-order sliding PSN, whose neuronal dynamics are

$$H[t] = \sum_{i=0}^{k-1} W_i \cdot X[t - k + 1 + i], \tag{14}$$

$$S[t] = \Theta(H[t] - V_{th}), \tag{15}$$

where $\mathbf{W} = [W_0, W_1, ..., W_{k-1}] \in \mathbb{R}^k$ is the learnable weight, $X[j] = 0$ for all $j < 0$, and $V_{th}$ is the learnable threshold. Similar to the masked PSN, $H[t]$ of the sliding PSN depends only on the latest $k$ inputs as well. More specifically, the weight slides over inputs and generates hidden states, which is a standard 1D convolution operation. Additionally, it can also be implemented by matrix-matrix multiplication. When the input sequence $\mathbf{X} \in \mathbb{R}^{T \times N}$ arrives and its length $T$ is known, the matrix $\mathbf{A} \in \mathbb{R}^{T \times T}$ can be generated as

$$\mathbf{A}[i][j] = \begin{cases} W_{k-1-i+j}, & i+1-k \leq j \leq i \\ 0, \text{otherwise} \end{cases}, \tag{16}$$

then $\mathbf{H} = \mathbf{A}\mathbf{X}$. According to the results of our experiments, using matrix-matrix multiplication is faster than using convolution.

### 3.5 Summary of the PSN Family

As a summary, Fig.2 shows the comparison of the parameter number $n_{param}$ and the generation of hidden states of the PSN family. Remarkably, using the PSN in deep SNNs will add a negligible number of parameters because $T$ is small in directly trained SNNs. For example, using the PSN with $T = 4$ will add 340 and 200 parameters in spiking ResNet-18 and VGG-11, which only increase $0.00291\%$ and $0.00015\%$ parameters of the original SNNs. Additionally, the PSN family has only one hidden state $\mathbf{H}$, making them consume less memory than the vanilla spiking neurons with at least two hidden states $\mathbf{H}, \mathbf{V}$ during training, which are analyzed and verified by experiments in the supplementary materials.

## 4 Experiments

In this section, we report the experiment results of evaluating the PSN family in aspects of simulation performance and temporal/static data classifying accuracy. All experiments are based on the SpikingJelly [52] framework. The details of the training are provided in the supplementary materials.

### 4.1 Simulation Speed Benchmark

We evaluated the simulation speed benchmark of the PSN and the LIF neuron, which is widely used in deep SNNs and acts as the baseline of the vanilla spiking neurons. Note that the implementations and simulation speed of the masked PSN and the sliding PSN are almost identical to the PSN. Thus, we take the PSN as the benchmark for the PSN family. The benchmark is running in two modes, which are inference and training. In inference, we use the PyTorch just-in-time (JIT) compiling to fuse Eq.(1)-(3) of the LIF neuron across all time-steps into a single function to accelerate by avoiding the calling overhead of too many tiny CUDA kernels. However, PyTorch JIT does not support modifying the backward for a fused forward function, so the surrogate method used in the backward of Eq.(2)

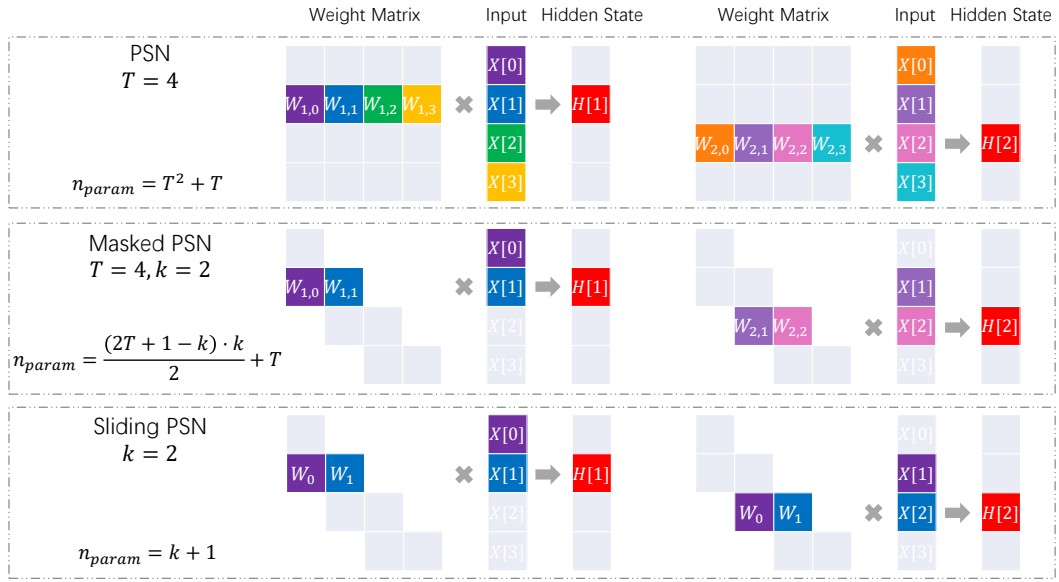

Figure 2: Comparison of the PSN family on the parameter number $n_{param}$ and the generation of hidden states.

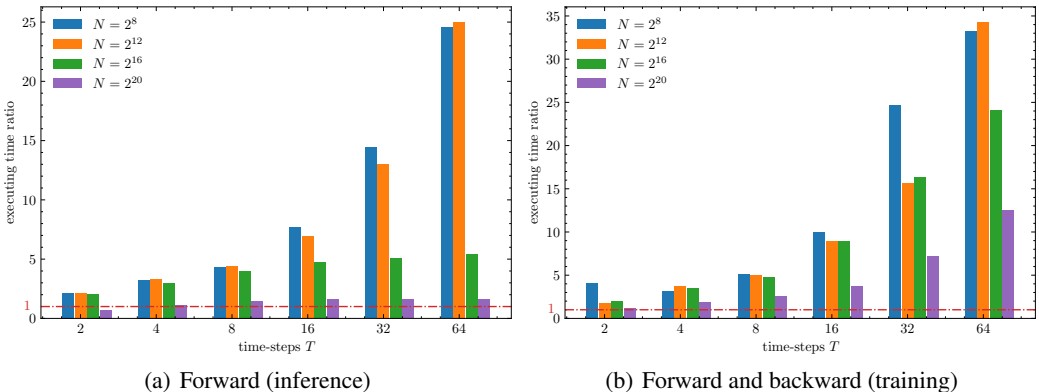

(a) Forward (inference)  (b) Forward and backward (training)

Figure 3: The executing time ratio of the LIF neuron and the PSN for one forward in inference and one forward and backward in training.

breaks the fusion of all three equations in training. Thus, we have to use three smaller JIT functions at each time-step for the LIF neuron in training. For the PSN, there is no difference between the implementations for inference and training, which both involve a matrix-matrix multiplication and an element-wise operation. We tested the neuron numbers $N = 2^8, 2^{12}, 2^{16}, 2^{20}$ and the time-step numbers $T = 2, 4, 8, 16, 32, 64$, which are typical options for deep SNNs. Denote the executing time of two neurons as $t_{PSN}$ and $t_{LIF}$ respectively, then the ratio $\frac{t_{LIF}}{t_{PSN}}$ is shown in Fig.3. It can be found that in most cases, the simulation of the PSN is much faster than that of the LIF neuron.

## 4.2 Learning Long-term Dependencies

To verify the learning ability for long-term dependencies of different spiking neurons, we evaluated their performance on classifying sequential CIFAR10 and CIFAR100. In these tasks, the image will be sent to the SNN column by column, which is similar to how humans read from left to right. The sequential image classification task is widely used to evaluate the learning ability of a network for long-term dependencies. The network structure is modified from [15] by replacing 2D convolutional/max pooling layers with 1D convolutional and average pooling layers, and removing

| Dataset \ Neuron | PSN | masked PSN | sliding PSN | GLIF[16] | KLIF[18] | PLIF[15] | LIF | LIF wo reset |
|---|---|---|---|---|---|---|---|---|
| Sequential CIFAR10 | 88.45 | 85.81 | 86.70 | 83.66 | 83.26 | 83.49 | 81.50 | 79.50 |
| Sequential CIFAR100 | 62.21 | 60.69 | 62.11 | 58.92 | 57.37 | 57.55 | 55.45 | 53.33 |

Table 1: The test accuracy (%) of different spiking neurons on sequential CIFAR.

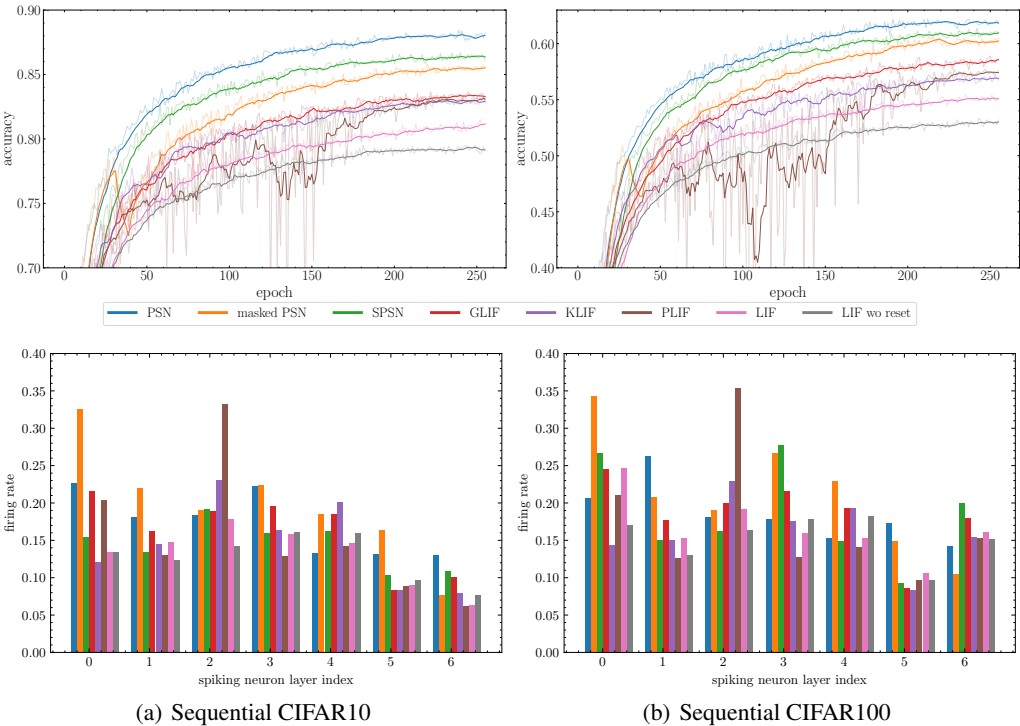

(a) Sequential CIFAR10  (b) Sequential CIFAR100

Figure 4: The accuracy curves and firing rates of different neurons for classifying sequential CIFAR.

the voting layers. Due to the column-by-column input, $T$ is 32, which is the width of the image. The compared neurons include PSN, 32-order masked PSN, 32-order sliding PSN, GLIF with channel-wise parameters, KLIF, PLIF, and the LIF neuron w/wo reset. We use the optimal reset options determined by ablation experiments for different neurons, which are using soft reset and detach reset [53] for the PLIF and LIF neurons, and using soft reset for the KLIF neuron. The GLIF neuron uses learnable gates to control all options, which do not need to be set manually. For the masked PSN, $\lambda = \min(1, 8 \cdot epoch/(epochs - 1))$, where $epoch$ denotes the current training epoch, and $epochs$ denotes the total number of epochs.

Tab.1 shows the accuracy of all neurons on sequential CIFAR. It can be found that the rank of accuracy is PSN > sliding PSN > masked PSN > GLIF > PLIF > KLIF > LIF > LIF wo reset, which is as expected. The PSN exploits information from all time-steps and has the highest learning ability, but it may be criticized for unfair comparison because it can read all columns at once. Therefore, the accuracy of the masked PSN and the sliding PSN is more convincing because they can still run in a step-by-step mode. Due to the introduction of high-order information, they work better than any other traditional spiking neurons. Meanwhile, the GLIF neuron is the best among traditional spiking neurons, indicating that fusing different kinds of charging, decaying, and resetting increases learning ability. The LIF neuron without reset has lower accuracy than the LIF neuron, indicating that the direct removal of reset drops the learning ability of the vanilla spiking neurons. Thus, using high-order information as the PSN family is necessary to fix this decline.

To make clear comparisons, we plotted the accuracy curves during training and firing rates of each spiking neuron layer of trained SNNs with different spiking neurons, as shown in Fig.4. These accuracy curves are the moving averages of 8 epochs, and the light-colored curves show the original accuracy. The curves of the PSN family are almost always above other curves, which clearly show their superior performance on convergence speed and final accuracy. Note that there is a sharp drop

| Dataset | Method | Spiking Network | Time-steps | Accuracy(%) |
|---|---|---|---|---|
| CIFAR10 | Dspike[38] | Modified ResNet-18 | 6 | 94.25 |
| | TET[39] | ResNet-19 | 6 | 94.50 |
| | TDBN[30] | ResNet-19 | 6 | 93.16 |
| | TEBN[33] | ResNet-19 | 6 | 94.71 |
| | PLIF[15] | PLIF Net | 8 | 93.50 |
| | KLIF[18] | Modified PLIF Net | 10 | 92.52 |
| | GLIF[16] | ResNet-19 | 6 | 95.03 |
| | PSN | Modified PLIF Net | 4 | 95.32 |
| ImageNet | Dspike[38] | ResNet-34 | 6 | 68.19 |
| | | VGG-16 | 5 | 71.24 |
| | TET[39] | SEW ResNet-34 | 4 | 68.00 |
| | TDBN[30] | ResNet-34 with double channels | 6 | 67.05 |
| | TEBN[33] | SEW ResNet-34 | 4 | 68.28 |
| | GLIF[16] | ResNet-34 | 4 | 67.52 |
| | SEW ResNet[40] | SEW ResNet-18 | 4 | 63.18 |
| | | SEW ResNet-34 | 4 | 67.04 |
| | PSN | SEW ResNet-18 | 4 | 67.63 |
| | | SEW ResNet-34 | 4 | 70.54 |
| CIFAR10-DVS | Dspike[38] | ResNet-18 | 10 | 75.40 |
| | TET[39] | VGG | 10 | 83.17 |
| | TDBN[30] | ResNet-19 | 10 | 67.80 |
| | TEBN[33] | VGG | 10 | 84.90 |
| | PLIF[15] | PLIF Net | 20 | 74.80 |
| | KLIF[18] | Modified PLIF Net | 15 | 70.90 |
| | GLIF[16] | Wide 7B Net | 16 | 78.10 |
| | SEW ResNet[40] | Wide 7B Net | 16 | 74.40 |
| | sliding PSN ($k = 2$) | VGG | 4, 8, 10 | 82.30, 85.30, 85.90 |

Table 2: Comparison of the PSN family and other methods.

in the curves of the masked PSN around epoch 32, which is caused by the mask becoming completely binary. The firing rates indicate that uninterrupted firing does not occur with the removal of neuronal resetting. In general, the PSN and the masked PSN are more easily to be activated and show higher firing rates than others. Meanwhile, the firing rates of the sliding PSN and the LIF neuron without reset are slightly higher than the vanilla spiking neurons. Thus, we conclude that removal of reset does increase firing rates, but the increment degree depends on the neuron's structure. On the other hand, considering the huge improvement in simulation efficiency and long-term dependencies learning ability, the slightly higher firing rate is acceptable.

Fig.5 shows the accuracy-order curve on the sequential CIFAR100 dataset for the masked PSN and sliding PSN, with the highest accuracy marked by a red ★. It can be seen that when the order increases from 1 to 4, the accuracy improves quickly. When we continue to increase the order, the improvement is marginal and the accuracy even drops. The highest accuracy is obtained by a large, but not the largest order, which is 20 and 31 for the masked PSN and sliding PSN, respectively. But in most cases, high accuracy can be guaranteed with $k = T$. Remarkably, the curve of sliding PSN is almost always above the curve of masked PSN, and excels that of the PSN when $k = 21$ and $k = 31$, indicating that the time-invariant parameters have better generalization ability to process temporal information.

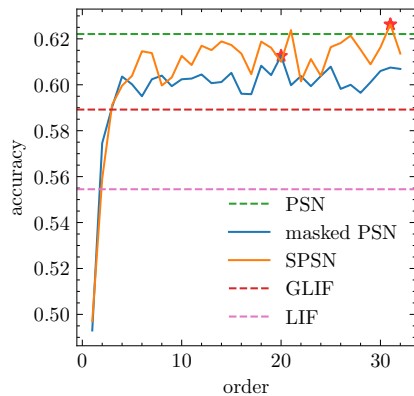

Figure 5: The accuracy-order curve on the sequential CIFAR100.

### 4.3 Static and Neuromorphic Data Classification

Static and neuromorphic data classification tasks are also common benchmarks for SNNs. We evaluated the PSN family on the static CIFAR10, ImageNet datasets, and the neuromorphic CIFAR10-DVS [54] dataset. The number of time-steps for static datasets is small, and the latency is not the main issue. Thus, we use the PSN in CIFAR10 and ImageNet. According to the results on sequential CIFAR, the sliding PSN uses fewer parameters but gets higher accuracy, and we will use it for CIFAR10-DVS. The results are listed in Tab.2.

**CIFAR10** We modified the network structure of [15], or the PLIF Net, by using average pooling and removing voting layers. Our PSN uses the shortest time-steps $T = 4$ and achieves the highest accuracy of 95.32%.

**ImageNet** We used the SEW ResNet [40] for classifying the ImageNet dataset. We verified the performance of the SPN on SEW ResNet-18/34 and got a stable 3+% accuracy improvement over the original SEW ResNet with the IF neuron. Our method achieves 70.54% accuracy and is only second to the Dspike method using one more time-step and VGG-16, which has a number of parameters 6.3 times as many as ours.

**CIFAR10-DVS** We used the VGG structure from [39]. Considering the fact that the DVS data contains temporal features and $T$ is larger than the static datasets, we use the 2-order sliding PSN. We achieve 82.3 % accuracy with $T = 4$, which is the first work to get 80+% accuracy in such few time-steps. When increasing time-steps to $T = 8$, we achieve 85.30% accuracy, which is 2+% higher than that of the previous SOTA method [33] and we use two fewer time-steps. With $T = 10$, the accuracy is further improved to 85.90%.

## 5    Conclusion

In this paper, we remove reset from neuronal dynamics of vanilla spiking neurons and propose the PSN family, including the PSN, the masked PSN, and the sliding PSN. The PSN family can be simulated in parallel, which greatly accelerates the training of deep SNNs. Weights of inputs in the PSN family can be fully connected, or masked/shared with a custom order. Experimental results of simulation speed and temporal/static data classification verify the efficiency and accuracy of the PSN family when compared with previous spiking neurons. Our work may be a milestone as well as a cornerstone for modern spiking neurons.

## Acknowledgments and Disclosure of Funding

This work is supported by grants from the National Natural Science Foundation of China (62027804, 61825101, 62088102, and 62176003), the major key project of the Peng Cheng Laboratory (PCL2021A13), and the Agence Nationale de la Recherche under Grant ANR-20-CE23-0004-04 DeepSee.

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
