# 6 Supplementary Material for *Parallel Spiking Neurons with High Efficiency and Ability to Learn Long-term Dependencies*

## 6.1 Environments for Simulation Speed Benchmark

The simulation speed benchmark is carried out on a Ubuntu 20.04.6 LTS server with the Intel Xeon(R) Gold 6226R CPU, 192G memory, and the NVIDIA Quadro RTX 6000 (24G) GPU. The PyTorch vision is 1.12.1, the CUDA version is 11.3.1, and the GPU driver version is 525.105.17.

## 6.2 Parameters Initialization

For the PSN and the masked PSN, the weights are initialized by the kaiming uniform [55]. More specifically, the values are initialized from $\mathcal{U}(-\sqrt{5}, \sqrt{5})$. For the sliding PSN, the weight is initialized by an exponential decay method

$$W_i = 2^{i-k+1}, \quad i = 0, 1, ..., k-1 \tag{17}$$

which is similar to the weight of the LIF neuron with $\tau_m = 2$ and removing reset. The thresholds of the PSN family are all initialized as $1$.

## 6.3 Implementation of the Sliding PSN

As mentioned in the main text, the sliding PSN can be implemented by both the matrix-matrix multiplication and the 1D convolution. However, the 1D convolution is much slower than the matrix-matrix multiplication, which is caused by:

1) The parallelism of convolution is lower than matrix-matrix multiplication, as we can find that the $1 \times 1$ convolution is often implemented by the fully connected layer in modern deep ANNs, such as ConvNeXt [56].

2) The 1D convolution of the sliding PSN requires the input with a shape of $(..., 1, T)$, while the default data format in SpikingJelly is $(T, N, ...)$. Thus, the transpose operation has to be used to move the time-step dimension, which copies data and slows down the speed.

## 6.4 Training Hyper-parameters

The main hyper-parameters for different datasets are shown in Tab.3. Unless otherwise specified, the weight decay (wd) is 0, the momentum is 0.9 for the SGD optimizer, the learning rate scheduler is the cosine annealing schedule [57], the automatic mixed precision training is used, and the surrogate function is the arctan surrogate function $\sigma(x) = \frac{\alpha}{2(1+(\frac{\pi}{2}\alpha x)^2)}$ with $\alpha = 4$ from [15]. Other training options are listed as follows.

| Dataset | Optimizer | Batch Size | Epoch | Learning Rate | GPUs | Loss |
|---|---|---|---|---|---|---|
| Sequential CIFAR10/100 | SGD 
 AdamW(sliding PSN) | 128 | 256 | 0.1 
 0.001(sliding PSN) | 1 | CE |
| CIFAR10 | SGD | 128 | 1024 | 0.1 | 1 | CE |
| ImageNet | SGD | 64 (SEW-18) 
 32 (SEW-34) | 320 | 0.1 | 8 | TET |
| CIFAR10-DVS | SGD, wd=5e-4 | 32 | 200 | 0.1 | 2 | TET |

Table 3: Training hyper-parameters for different datasets.

**Static/Sequential CIFAR** The transforms include random mixup [58] with $p = 1, \alpha = 0.2$, random cutmix [59] with $p = 1, \alpha = 1$, random choice of two mix methods with $p = 0.5$, random horizontal flip with $p = 0.5$, trivial augment [60], normalization, random erasing [61] with $p = 0.1$, and label smoothing [62] with the amount $0.1$. The number of channels is 256 for static CIFAR and 128 for sequential CIFAR.

**ImageNet** Transforms are identical to [40]. We loaded the pre-trained weights from the standard ResNet provided by the ANN community as better initialized parameters. Similar techniques have been widely used in surrogate training [63, 64, 65].

| $T$ | $N$ | $\mathcal{M}_{NO}$ | $\mathcal{M}_{IF}$ | $\mathcal{M}_{PSN}$ | $\Delta_{IF-NO}$ | $\Delta_{PSN-NO}$ | $\frac{\Delta_{IF-NO}}{\Delta_{PSN-NO}}$ | $\frac{\Delta_{IF-NO}-\Delta_{PSN-NO}}{T\cdot N}$ |
|----|----|---------|---------|---------|-------|-------|-----|------|
| 16 | 16 | 35129.9 | 70563.9 | 53675.9 | 35434 | 18546 | 1.9 | 66.0 |
| 8  | 16 | 18935.9 | 36863.9 | 28225.9 | 17928 | 9290  | 1.9 | 67.5 |
| 16 | 8  | 18933.9 | 36099.9 | 28223.9 | 17166 | 9290  | 1.8 | 61.5 |

Table 4: Comparison of the memory consumption of training SNNs with the IF neuron and the PSN.

**CIFAR10-DVS** Transforms are identical to [33]. We used the almost identical network structure, training hyper-parameters, and options as [33], except that we use the plain BN rather than TEBN.

## 6.5 Memory Consumption of the PSN family

The memory complexity of the SNN during training can be approximated as $\mathcal{O}(\mathcal{W} + T \cdot (\mathcal{X} + \mathcal{H}))$, where $\mathcal{W}$ is the number of synapses such as convolutional/linear layers and weights inside neurons such as $\boldsymbol{W}$ in the PSN, $T$ is the number of time-steps, $\mathcal{X}$ is the input/output of all layers at a single time-step, and $\mathcal{H}$ is the hidden state of all layers at a single time-step. When comparing the SNN using the vanilla spiking neurons or the PSN family, the difference in memory consumption mainly depends on $\mathcal{H}$.

For the vanilla spiking neurons, it can be found that the number of hidden states is at least 2, which are $\boldsymbol{H}$ and $\boldsymbol{V}$. The number of hidden states increases if the spiking neuron integrates more complex neuronal dynamics such as adaptative thresholds. While for the PSN, only $\boldsymbol{H}$ is used and the number of hidden states is 1. Thus, it can be found that the memory consumption of the PSN is $\mathcal{O}(T \cdot N)$ less than the vanilla spiking neurons, where $N$ is the number of neurons in the SNN.

We design an interesting experiment to verify our analysis. We measure $\mathcal{M}_{NO}$, which is the memory consumption of training the network with only synapses but no neurons to estimate $\mathcal{O}(\mathcal{W} + T \cdot \mathcal{X})$. Then we measure $\mathcal{M}_{IF}$ and $\mathcal{M}_{PSN}$, which are SNNs with the simplest vanilla spiking neuron, the IF neuron, and the PSN to estimate $\mathcal{O}(\mathcal{W}+T\cdot(\mathcal{X}+\mathcal{H}))$. Thus, we can estimate the memory consumption of hidden states in two SNNs by $\Delta_{IF-NO} = \mathcal{M}_{IF} - \mathcal{M}_{NO}$ and $\Delta_{PSN-NO} = \mathcal{M}_{PSN} - \mathcal{M}_{NO}$. We test on the VGG-11 structure, and the results are shown in Tab.4. It can be found that the $\Delta_{PSN-NO} < \Delta_{IF-NO}$, and $\frac{\Delta_{IF-NO}}{\Delta_{PSN-NO}} \approx 2$ in all cases, which matches our analysis. More specifically, the IF neuron uses 2 hidden states, and the PSN uses 1 hidden state, then the ratio of their extra memory consumption should be 2. It can also be found that $\frac{\Delta_{IF-NO}-\Delta_{PSN-NO}}{T\cdot N}$ is approximately a constant, which also corresponds to our analysis that the memory consumption of the PSN is $\mathcal{O}(T \cdot N)$ less than the IF neuron.