# OpenReview forum: "Parallel Spiking Neurons with High Efficiency and Ability to Learn Long-term Dependencies"
_NeurIPS.cc/2023/Conference — NeurIPS 2023 poster_

### Official Review · Reviewer_G2d4 · 2023-06-24

**Soundness:** 3 good
**Presentation:** 3 good
**Contribution:** 3 good
**Rating:** 6
**Confidence:** 4

**Summary:**

This paper introduces the parallel spiking neuron (PSN) and several variants. The primary benefit of PSN over the existing spiking neurons is the parallel implementation on digital hardware, which brings dozens of times of acceleration on GPU. The accuracy also demonstrates the effectiveness of this paper.

**Strengths:**

1. The idea of parallelized neuron implementation is interesting and can be efficiently processed on GPU.

2. The experiments show good performances.

**Weaknesses:**

The authors pinpoint a disadvantage of current SNN, that is, low running speed on GPUs. However, it is known that GPU is not the most ideal device for deploying SNN, rather, it should be the neuromorphic hardware. PSN focuses on the optimization of SNN on GPU, however, GPU cannot utilize the binary spikes to lower the energy. So, even if PSN can accelerate the inference on GPU, compared to ANN on GPU, there is no advantage of efficiency still compared to ANNs. I was wondering whether the optimization on GPU is really useful cause people can always use ANNs on GPU.



**Questions:**

Can authors show the ablation study of PSN and vanilla SNN on static image datasets?


**Limitations:**

Listed above.

---

> ### Author Rebuttal · Authors · 2023-08-08
>
> Thanks for your encouraging comments about the acceleration and accuracy of the PSN family. Our responses to the weakness are as follows.
>
> > I was wondering whether the optimization on GPU is really useful cause people can always use ANNs on GPU.
>
> A typical workflow to use SNNs is:
>
> 1. Train SNNs to get high task performance
> 2. Prune and quantize the weights of SNNs during or after training
> 3. Deploy SNNs to neuromorphic chips
>
> For the moment, many researches aim at steps 1 and 2. Although GPU is not the target device for SNNs to deploy, the massively parallel computing ability makes it become the most widely used devices to train SNNs. We believe that the introduction of PSN decreases the training cost and benefits the SNN community.
>
> Meanwhile, although the PSN family has high efficiency in GPUs, it is also compatible with neuromorphic chips. Please refer to our response to the question "How it could map to the hardware?" from reviewer P1JU for more details.
>
> > Can authors show the ablation study of PSN and vanilla SNN on static image datasets?
>
> Thanks for your suggestion. We add ablation experiments on CIFAR10. We use the same network from the SNN to classify CIFAR10 in Table 2. Due to the limited rebuttal period, we only use 128 channels for the SNN and train 128 epochs. The results are shown in Table R9. Note that $T=4$ and we also train the masked PSN and the sliding PSN with $k=0,1,2,3$.
>
> | Neuron          | IF    | LIF   | PLIF  | KLIF  | GLIF  | PSN   |
> | --------------- | ----- | ----- | ----- | ----- | ----- | ----- |
> | **Accuracy(%)** | 92.24 | 91.86 | 92.24 | 92.47 | 91.90 | 92.95 |
>
> | Neuron\Order | 1     | 2     | 3     | 4     |
> | ------------ | ----- | ----- | ----- | ----- |
> | Masked PSN   | 92.59 | 92.44 | 92.75 | 91.94 |
> | Sliding PSN  | 90.81 | 91.79 | 92.13 | 92.46 |
>
> **Table R9.  Ablation experiments using different spiking neurons on CIFAR10.**
>
> As [1] suggests, the spiking neurons without leaks may have better accuracy in static datasets. And we also add the experiment using the IF neuron. Table R10 shows that the accuracy rank on CIFAR10 is `PSN > Masked PSN (k=3) > KLIF > Sliding PSN (k=4) > PLIF = IF > GLIF > LIF`. The rank indicates that the PSN family also performs better than most of vanilla spiking neurons in static datasets.
>
> ```
> [1] Fang, Wei, et al. "Deep residual learning in spiking neural networks." Advances in Neural Information Processing Systems 34 (2021): 21056-21069.
> ```

---

### Official Review · Reviewer_Xbwd · 2023-07-05

**Soundness:** 4 excellent
**Presentation:** 3 good
**Contribution:** 3 good
**Rating:** 7
**Confidence:** 4

**Summary:**

The paper presents an approach to improve the efficiency and accuracy of Spiking Neural Networks by using a dependency method to generate hidden states, resulting in parallelizable neuronal dynamics and a significant increase in simulation speed.

**Strengths:**

The authors analyze the impact of removing the reset function from standard charge-fire-reset neuronal dynamics, proving that the parallelization of the spiking neuron can be achieved without it. They also present a general formula by rewriting the neuronal dynamics without a reset and introduce the PSN, a spiking neuron with entirely parallelizable neuronal dynamics. The paper also assesses the PS family's performance on sequential, static, and neuromorphic data classification tasks, showing that they attain higher accuracy than traditional spiking neurons.

**Weaknesses:**

Please see Questions.

**Questions:**

1. Since the proposed models use parallel input of X[t-1], X[t], X[t+1], I am wondering how memory-intensive the models will be for parallel processing, and especially how it changes for the different datasets. Basically, it would be interesting to see a comparison of FLOPS/MACs for the standard vanilla spiking neurons and the proposed parallel spiking neurons

2. The VGG shows better performance for ImageNet and CIFAR-DVS. The authors show the performance of the VGG model is currently state of the art using vanilla neurons for the ImageNet dataset. It would be great if the authors could report the results of the VGG model with parallel neurons for the ImageNet dataset. I understand the tight time constraint and hope this won't be very difficult, especially because the authors have already shown results for VGG on the CIFAR-10 DVS

3. I am also curious how stable are these parallel spiking neurons compared to the vanilla neurons since the reset is removed. In the presence of input/weight noise, will it still perform as well?

4. When the authors evaluate the long-term dependencies, it would be interesting to see results on non-image datasets in Supplementary. Mainly because there is a high degree of correlation among the pixels for such image datasets, and I was just wondering how well it would work for other non-image-based datasets.



**Limitations:**

It would be great if the authors could include what are the key limitations of this work and what are the trade-offs with the current vanilla spiking neurons.

---

> ### Author Rebuttal · Authors · 2023-08-08
>
> Thanks for your positive comments. Please refer to "To All Reviewers" for our discussions about the trade-off with vanilla spiking neurons. Our responses to your constructive questions are as follows.
>
> ## **Question 1**
>
> Thanks for your variable question. We have summarized the number of memory reading/writing and FLOPS of spiking neurons in Table R5.
>
> |             | Memory Readings                   | Memory Writings | FLOPs              |
> | ----------- | --------------------------------- | --------------- | ------------------ |
> | LIF/PLIF    | $5T$                              | $3T$            | $9T$               |
> | KLIF        | $6T$                              | $3T$            | $10T$              |
> | GLIF        | $11T$                             | $2T$            | $20T$              |
> | PSN         | $T^{2} + 2T$                      | $T$             | $2T^{2}$           |
> | Masked PSN  | $\frac{(2T+1-k) \cdot k}{2} + 2T$ | $T$             | $(2T+1-k) \cdot k$ |
> | Sliding PSN | $k+1+T$                           | $T$             | $(2T+1-k) \cdot k$ |
>
> **Table R5. Counting of the memory reading/writing and FLOPs of different neurons.**
>
> ## **Question 2**
>
> Considering the fact that the tuning of hyper-parameters also requires many comparative experiments, we are not able to finish the training of VGG-16 during the rebuttal period.
>
> ## **Question 3**
>
> Thanks for your suggestion. We add experiments about input/weight noise on the sequential CIFAR100 classification task. Note that all noise is added during both training and inference.
>
> #### Noised Input
>
> We add a Gaussian noise with mean 0 and variance $\sigma^2$ on inputs $X$. The results are shown in Table R6. It can be found that at each noise level, the PSN family is still better than the LIF neuron.
>
> | Neuron\\$\sigma$ | 0 (no noise) | 0.1   | 0.2   | 0.3   |
> | ---------------- | ------------ | ----- | ----- | ----- |
> | LIF              | 55.45        | 54.01 | 51.53 | 49.17 |
> | PSN              | 62.21        | 61.31 | 59.21 | 56.88 |
> | Masked PSN       | 60.69        | 57.63 | 56.17 | 56.76 |
> | Sliding PSN      | 62.11        | 59.72 | 56.74 | 55.17 |
>
> **Table R6: The accuracy on sequential CIFAR100 with noised inputs**.
>
> #### Noised Weights
>
> We add a Gaussian noise with mean 0 and variance $\sigma^2$ on weights $W$ and bias $B$ (if have) of all convolutional, batch normalization, and fully connected layers.
>
> | Neuron\\$\sigma$ | 0 (no noise) | 0.001 | 0.005 | 0.0075 | 0.01  |
> | ---------------- | ------------ | ----- | ----- | ------ | ----- |
> | LIF              | 55.45        | 52.26 | 31.79 | 21.48  | 14.98 |
> | PSN              | 62.21        | 57.56 | 20.56 | 12.49  | 4.67  |
> | Masked PSN       | 60.69        | 53.58 | 23.16 | 11.79  | 4.58  |
> | Sliding PSN      | 62.11        | 58.05 | 38.81 | 29.33  | 21.27 |
>
> **Table R7: The accuracy on sequential CIFAR100 with noised weights**.
>
> The results are shown in Table R7. When noise becomes large, the accuracy of both the LIF neuron and the PSN family drops quickly. The accuracy rank in the large noise environment is `Sliding PSN > LIF > PSN > Masked PSN`. It is worth noting that both the sliding PSN and the LIF neuron use shared weights across time-steps, while the PSN and the sliding PSN use temporal-wise weights. The former might have better robustness for noised network weights.
>
> ## **Question 4**
>
> Thanks for your advice. We add two kinds of experiments beyond image classification.
>
> ### Reinforcement Learning and Control
>
> We evaluate the performance of SNNs with different spiking neurons on the reinforcement learning and control tasks. We evaluate four typical tasks from OpenAI Gym with non-image inputs, which include Ant-v3, HalfCheetah-v3, Hopper-v3, and Walker2d-v3. We use similar encoding and decoding methods from [1]. The results are reported in Table R8. It can be found that the performance of the PSN and the sliding PSN is much higher than other neurons.
>
> |                                | Ant-v3 |         | HalfCheetah-v3 |         | Hopper-v3 |         | Walker2d-v3 |         | Average Performance Ratios |
> | ------------------------------ | ------ | ------- | -------------- | ------- | --------- | ------- | ----------- | ------- | -------------------------- |
> | DAN [2]                        | 5472   | 100.00% | 10471          | 100.00% | 3520      | 100.00% | 4999        | 100.00% | 100.00%                    |
> | PopSAN (current-based LIF [1]) | 4848   | 88.60%  | 10523          | 100.50% | 517       | 14.69%  | 4199        | 84.00%  | 71.94%                     |
> | PopSAN (LIF)                   | 4991   | 91.21%  | 8500           | 81.18%  | 2613      | 74.23%  | 3751        | 75.04%  | 80.41%                     |
> | PopSAN (PSN)                   | 5210   | 95.21%  | 9622           | 91.89%  | 3255      | 92.47%  | 4248        | 84.98%  | 91.14%                     |
> | PopSAN (SlidingPSN)            | 5362   | 97.99%  | 9849           | 94.06%  | 3406      | 96.76%  | 4603        | 92.08%  | 95.22%                     |
>
> **Table R8: The performance comparison of spiking neurons on reinforcement learning and control tasks.**
>
> ### Speech Recognition
>
> We add experiments about speech recognition. We use the same network structure and data processing methods from [3]. We achieve 95.56% accuracy, which is higher than the LIF neuron with 94.5% in [3].
>
> ```
> [1] Tang, Guangzhi, et al. "Deep reinforcement learning with population-coded spiking neural network for continuous control." Conference on Robot Learning. PMLR, 2021.
> [2] Fujimoto S, Hoof H, Meger D. Addressing function approximation error in actor-critic methods[C]//International conference on machine learning. PMLR, 2018: 1587-1596.
> [3] Pellegrini, Thomas, Romain Zimmer, and Timothée Masquelier. "Low-activity supervised convolutional spiking neural networks applied to speech commands recognition." 2021 IEEE Spoken Language Technology Workshop (SLT). IEEE, 2021.
> ```

---

> > ### Comment · Reviewer_Xbwd · 2023-08-18
> > **Reply to the Author's Response**
> >
> > I thank the authors for their detailed response and for clarifying my doubts.

---

### Official Review · Reviewer_P1JU · 2023-07-05

**Soundness:** 4 excellent
**Presentation:** 3 good
**Contribution:** 3 good
**Rating:** 6
**Confidence:** 5

**Summary:**

This paper proposes the Parallel Spiking Neuron (PSN), which generates hidden states that are independent of their predecessors, resulting in parallelizable neuronal dynamics and extremely high simulation speed. The weights of inputs in the PSN are fully connected, which maximizes the utilization of temporal information. The authors evaluate  the PSN family on simulation speed and temporal/static data classification, and  the results show the overwhelming advantage of the PSN family in efficiency and  accuracy.

**Strengths:**

1. The motivation considering the parallel spiking neurons for high simulation speed is interesting and important for the futural application of spiking neural networks.
2. The experiments on the large datasets, such as ImageNet, improves the techinical soundness of the spiking neural networks.

**Weaknesses:**

1. Since the main goal of the proposed parallel spiking neuron model is to improve the simulation speed, the current experiments do not reflect that advantage over other methods, in which only focusing on the accuracy and the firing rates analysis.
2. The network architecture of the spiking networks in Table 2 is not clear.

**Questions:**

1. Please explain the advantage of the parallel spiking neurons as mentioned "high simulation speed " in abstract.
2. How about the whole training and test time of the proposed SNN compared with other methods?
3. Is there any limitation once applied the parallel spiking neurons into the neuromorphic hardware？How it could map to the hardware?
4.What is the network architecture used in Table2, such as the "VGG"? Please describe the detailed network architecture in the Table 2.

**Limitations:**

1. Since the authors claim that the main goal of the proposed parallel spiking neuron model is to improve the simulation speed, the current experiments do not reflect that advantage over other methods, in which only focusing on the accuracy and the firing rates analysis.
2. The network architecture of the spiking networks in Table 2 is not clear.

The above limitations have been explained clearly in the following response.

---

> ### Author Rebuttal · Authors · 2023-08-08
>
> Thanks for your comprehensive comments. Please refer to "To All Reviewers" for responses to **Question 3**. Other point-to-point responses are as follows.
>
> ## **Weaknesses 1 and Questions 1, 2**
>
> > Please explain the advantage of the parallel spiking neurons as mentioned "high simulation speed " in abstract.
>
> You can refer to Section 4.1 and Figure 3 for experimental results about the training and inference speed.
>
> > How about the whole training and test time of the proposed SNN compared with other methods?
>
> Comparable methods include PLIF, KLIF, and GLIF. These neurons are more complex than the LIF neuron with additional operations. Thus, we add experiments about speed comparison between the PSN and the LIF neuron, and if the PSN is faster than the LIF neuron, then it is also faster than other more complex neurons.
>
> We compare the training and inference speed (ms/batch) of the PSN and the LIF neuron with the SEW ResNet ResNet-18/34 and $T=4, 8$, which is the typical option for training deep SNNs. We use the LIF neuron from SpikingJelly, which provides the fastest implementation. The experiments are performed on an Ubuntu 18.04 server with an Intel Xeon Silver 4210R CPU, an NVIDIA A100-SXM-80GB GPU, and 256 GB of memory.
>
> |           | SEW ResNet-18 (T=4) |           | SEW ResNet-18 (T=8) |           | SEW ResNet-34 (T=4) |           | SEW ResNet-34 (T=8) |           |
> | --------- | ------------------- | --------- | ------------------- | --------- | ------------------- | --------- | ------------------- | --------- |
> |           | Train               | Inference | Train               | Inference | Train               | Inference | Train               | Inference |
> | PSN       | 45.94               | 10.91     | 47.99               | 19.17     | 60.63               | 16.74     | 71.52               | 30.87     |
> | LIF       | 60.40               | 10.55     | 98.10               | 18.82     | 94.09               | 17.52     | 149.86              | 30.42     |
> | LIF(cupy) | 49.71               | 12.08     | 53.81               | 18.74     | 77.49               | 17.36     | 82.38               | 30.19     |
>
> **Table R4. Comparison of speed (ms/batch) between the SNN using PSN and the LIF neuron.**
>
> The results are shown in Table R4. `LIF` is the LIF neuron implemented by PyTorch, and`LIF(cupy)` is the LIF neuron implemented by CuPy. The results show that the SNN using PSN is much faster than using the LIF neuron in training, and the advantage increases when the scale of the network is larger or $T$ increases. In inference, there is not much difference among speeds, which may be caused by that the speed of spiking neuron layers is not the bottleneck.
>
> Compared with `LIF(cupy)`, the speed advantage of PSN is not so significant with `LIF`. Note that `LIF(cupy)` fuses all operations concerning all time-steps into one single CUDA kernel, which minimizes the calling overhead of CUDA kernels including memory access time and kernel launch time. The PSN is implemented by PyTorch, and its operations include a matrix-matrix multiplication ($WX$), an element-wise subtraction ($H - B$), and a Heaviside function ($\Theta(H - B)$) in forward or a surrogate function ($\sigma (H - B)$) composed of many math operations in backward, which has higher calling overhead than `LIF(cupy)` using a single CUDA kernel. The speed of PSN can be improved further if its neuronal dynamics apart from the matrix-matrix multiplication are fused into one single CUDA kernel. However, considering the fact that PSN implemented by PyTorch is fast enough, we have not implemented the CuPy backend for PSN.
>
> ## Weaknesses 2 and Question 4
>
> > What is the network architecture used in Table2, such as the "VGG"? Please describe the detailed network architecture in the Table 2.
>
> Sorry for the unclear expression.
>
> Then the detailed network structure in Table 2 is:
>
> - Modified PLIF Net: `{{c256k3s1-BN-PSN}\*2-APk2s2}\*2-Flatten-FC4096-PSN-FC10`
> - SEW ResNet: the standard SEW ResNet [1] using PSN as spiking neuron layers
> - VGG: `{c64k3s1-BN-SPSN}-{c128k3s1-BN-SPSN}-APk2s2-{c256k3s1-BN-SPSN}*2-APk2s2-{{c512k3s1-BN-SPSN}*2-APk2s2}*2-Flatten-DP-FC10`, and `SPSN` is the sliding PSN with $k=2$
>
> ```
> [1] Fang, Wei, et al. "Deep residual learning in spiking neural networks." Advances in Neural Information Processing Systems 34 (2021): 21056-21069.
> ```

---

> > ### Comment · Reviewer_P1JU · 2023-08-11
> >
> > Thanks for the clarifications and the additional experiments. It addresses my concerns sufficiently.

---

### Official Review · Reviewer_sjmV · 2023-07-05

**Soundness:** 3 good
**Presentation:** 3 good
**Contribution:** 3 good
**Rating:** 7
**Confidence:** 4

**Summary:**

This paper removes the reset mechanism from the dynamics of conventional LIF/IF neurons, and proposes to reformulate the neuronal dynamics using matrix multiplication instead of the iterative updating for the membrane potential. This matrix multiplication then can be simulated in parallel to accelerate the training of deep SNNs.
This paper makes strong assumption that hidden states are independent of their predecessors, and proposes the Parallel Spiking Neuron (PSN) and its variants, masked PSN, sliding PSN.
The whole framework is built on unfolding the computing graph over the latency $T$ and then uses a fully connected layer $H = W X$ to replace the neuron dynamics.


**Strengths:**

1. Parallelization: Parallelizing in spiking neurons is an interesting topic in SNNs. By removing the reset mechanism, the neuronal dynamics can be reformulated in a non-iterative form. The proposed Parallel Spiking Neuron (PSN) framework allows for parallelized neuronal dynamics, enabling efficient computations across multiple processing units or threads.

2. Utilization of Temporal Information: The PSN utilizes fully connected weights for inputs, maximizing the utilization of temporal information and potentially enhancing the model's ability to capture temporal patterns.

3. High Simulation Speed: The PSN framework, with its independent hidden states and parallelizable dynamics, achieves extremely high simulation speed, which can be advantageous for real-time applications and large-scale simulations.


**Weaknesses:**

1. Lack of Reset Mechanism: The removal of the reset mechanism in the PSN may limit its ability to handle certain types of dynamics or tasks that rely on precise timing and reset behavior. The authors did not provide reasonable explanation of why neuronal resetting can be ignored, what should be done to compensate for reset removal.

2. Large number of trainable parameters introduced by the new Weights: The use of matrix multiplication in the PSN introduces additional trainable parameters in the new weights, the performance improvement could benefit from using more trainable parameters, but not the new PSN model. With the same number of parameters for both PSN and LIF, will the performance still perform good using the new PSN compared to LIF?

3. Using future information: when calculating $H[t] = \sum_{i=1}^T W_{t,i} X[i]$, in this PSN the future information is used. For the sliding PSN and masked PSN, the authors tried to avoid using future information. So that in order avoiding using the future information, it's better to use masked PSN and sliding PSN, but in the experiments, it's not clear which PSN version is used.

4. This paper makes strong assumption that hidden states are independent of their predecessors, the Parallel Spiking Neuron (PSN) is proposed based on this.  How to describe the neuron dynamics along the time-dimension if we remove this hidden state dependency?



**Questions:**

As above in Weakness.

The whole PSN framework is built on the condition $u(t) < V_{th}$, how to get spikes if this is the precondition for PSN?

The lack of dependency between successive time-steps is a concern for me. The whole framework is built on unfolding the computing graph over the latency $T$ and then uses a fully connected layer $H = W X$ to replace the neuron dynamics.  It doesn't make sense. Can you give more detailed explanation on this?

**Limitations:**


1. The whole PSN framework is built on the condition $u(t) < V_{th}$, not clear how to get spikes if this is the precondition for PSN.

2. Using future information: when calculating $H[t] = \sum_{i=1}^T W_{t,i} X[i]$, in this PSN the future information is used. For the sliding PSN and masked PSN, the authors tried to avoid using future information. So that in order avoiding using the future information, it's better to use masked PSN and sliding PSN, but in the experiments, it's not clear which PSN version is used.

3. This paper makes strong assumption that hidden states are independent of their predecessors, the Parallel Spiking Neuron (PSN) is proposed based on this.  How to describe the neuron dynamics along the time-dimension if we remove this hidden state dependency?

---

> ### Author Rebuttal · Authors · 2023-08-08
>
> Thanks for your constructive comments and questions, which are also helpful for other reviewers. Please refer to "To All Reviewers" for responses to **Weaknesses 1: Lack of Reset Mechanism**. Responses to other comments are as follows.
>
> ## **Weaknesses 2**
>
> Although the PSN family uses extra trainable parameters, the increasing memory cost is negligible. To explain this point clearly, we list the number of parameters of different layers, which are shown in Table R1. Note that the parameter number of masked PSN is  $T^{2} + T$ in training, and parameters are masked progressively. After training, the parameter number reduces to $\frac{(2T+1-k) \cdot k}{2} + T$ in inference.
>
> | Layer                 | Description                                                  | Params                                   |
> | --------------------- | ------------------------------------------------------------ | ---------------------------------------- |
> | convolutional layer   | $C_{in}$ input channels, $C_{out}$ output channels, $k_{h} \cdot k_{w}$ weight shape | $C_{in} \cdot C_{out} \cdot k_{h} k_{w}$ |
> | fully connected layer | $F_{in}$ input features, $F_{out}$ output features           | $F_{in} \cdot F_{out}$                   |
> | PSN                   | $T$ time-steps                                               | $T^{2} + T$                              |
> | Masked PSN            | $T$ time-steps, $k$ orders                                   | $T^{2} + T$                              |
> | Sliding PSN           | $T$ time-steps, $k$ orders                                   | $k+1$                                    |
>
> **Table R1. The number of parameters of different layers.**
>
> In lines 207-209, we have shown that using the PSN in deep SNNs will cause a negligible increase in parameters. Now let us take the SNNs in Table 1 as a new example. Denote the number of the SNN using the LIF neuron $P(LIF)$ as the baseline, we report the ratios of the number of parameters $\frac{P(neu)}{P(LIF)}$ of SNNs in Table 1. The results are shown in Table R2. Although $T=32$ is large in these SNNs, the results in Table 1 show that the using of the PSN family will increase no more than 5% extra parameters.
>
> | Dataset\Neuron          | PSN      | Masked PSN | SPSN     | GLIF     | KLIF     | PLIF     | LIF  | LIF wo reset |
> | ----------------------- | -------- | ---------- | -------- | -------- | -------- | -------- | ---- | ------------ |
> | **Sequential CIFAR10**  | 1.014398 | 1.014398   | 1.00045  | 1.077785 | 1.000014 | 1.000014 | 1    | 1            |
> | **Sequential CIFAR100** | 1.013777 | 1.013777   | 1.000431 | 1.074431 | 1.000013 | 1.000013 | 1    | 1            |
>
> **Table R2. The ratios of the number of parameters $\frac{P(neu)}{P(LIF)}$ of SNNs in Table 1.**
>
> > With the same number of parameters for both PSN and LIF, will the performance still perform good using the new PSN compared to LIF?
>
> We can easily build SNNs with fewer parameters. Denote the SNN using the LIF neuron for classifying the sequential CIFAR100 in Section 4.2 as the baseline. We reduce the number of the output/input channels of the first/second convolutional layer by 1, causing a slight reduction of parameters. We train these SNNs with the same hyper-parameters and training options as Section 4.2, and the results are shown in Table R3. The results indicate that the PSN family still has higher accuracy even with fewer parameters than the SNN using the LIF neuron.
>
> |                 | LIF (baseline) | PSN    | Masked PSN | Sliding PSN |
> | --------------- | -------------- | ------ | ---------- | ----------- |
> | **Params**      | 536548         | 520415 | 520415     | 513254      |
> | **Accuracy(%)** | 55.45          | 62.74  | 60.23      | 61.75       |
>
> **Table R3. Comparison of sequential CIFAR100 between LIF and PSN family with fewer parameters.**
>
> ## **Weaknesses 3**
>
> > but in the experiments, it's not clear which PSN version is used.
>
> You can refer to Table 2 and Section 4.3 for more details. In a word, we use the PSN for CIFAR10 and ImageNet, and the sliding PSN with order $k=2$ for CIFAR10-DVS.
>
> ## **Weaknesses 4**
>
> > How to describe the neuron dynamics along the time-dimension if we remove this hidden state dependency?
>
> You can refer to Section 3.1 for more details. Here we summarize that the iterative neuronal dynamics with removing neuronal reset can be reformulated to non-iterative equations, as Eqs. (5) and (7) show.
>
> ## Questions
>
> > The whole PSN framework is built on the condition $u(t)<V_{th}$, not clear how to get spikes if this is the precondition for PSN.
>
> We regret the unclear expression, which may mislead you. In line 151, we assume $H[t] < V_{th}$ as a method to ignore neuronal resetting. In lines 152-158, we claim that this method is meaningless, and a better method to ignore neuronal resetting is removing resetting from neuronal dynamics directly. In the PSN family, $H[t]$ is not restricted to be lower than the threshold $V_{th}$.
>
> > The lack of dependency between successive time-steps is a concern for me.
>
> Denote the initial value of $H[t]$ as $H[-1]$. In the vanilla spiking neuron, the neuronal dynamics is a typical Markov chain with the transfer function $g$ determined by the neuronal dynamics, and
> $$
> H[t] = g(H[t-1], X[t]) = g(g(H[t-2], X[t-1]), X[t]) = ...=g(g(g(H[-1], X[0]), X[1])..., X[t-1]).
> $$
> Thus, $H[t]$ is actually determined by $X[0], X[1], ..., X[t]$. In PSN, masked PSN and sliding PSN, $H[t]$ is determined by $X[0], X[1], ... ,X[T-1]$, $X[0], X[1], ..., X[t]$ and $X[t-k+1], X[t-k+2], ..., X[t]$.
> In conclusion, we can find that $H[t]$ in both vanilla spiking neurons and the PSN family is determined by input at specific time-steps. The main difference is that the dependency in vanilla spiking is indirect, which is implemented by a Markov chain. While in the PSN family, the dependency is direct, which is the weighted sum.

---

> > ### Comment · Reviewer_sjmV · 2023-08-16
> > **Response to authors**
> >
> >
> > I would like to thank the authors for providing thorough responses and for their efforts to enhance the paper based on my feedback. The clear and accurate explanations you provided exceeded my expectations, and I appreciate your dedication in addressing my questions.

---

### Author Rebuttal · Authors · 2023-08-08

Thanks for all reviewers' valuable comments. We are encouraged that reviewers find the idea of parallelizing spiking neurons interesting and commend the fast simulation speed of the PSN family. Meanwhile, most reviewers are concerned about the hardware implementation of the PSN family and the trade-off with vanilla spiking neurons. Our responses to these questions are as follows.

## Hardware Compatibility

To the best of our knowledge, the behavior of most event-driven neuromorphic chips is closer to the step-by-step forward propagation (send input $X[t]$ to the SNN and get output $Y[t]$) that each event is processed and routed between cores. Thus, the masked PSN and the sliding PSN can be implemented in these chips because they support the step-by-step forward propagation. The PSN requires input at all time-steps and can not work in these chips. While some heterogeneous chips [1, 2] support the layer-by-layer forward propagation (send inputs $X[0], X[1], ..., X[T-1]$ to the SNN and get outputs $Y[0], Y[1], ..., Y[T-1]$), and the PSN has the potential to work in this type of neuromorphic chips.
When deploying the PSN family on hardware, memory consumption should also be considered. Please refer to Figure 2 in the main text for details about the memory consumption of parameters. Additionally, the masked PSN and the sliding PSN require a stack with length $k$ to store $X[t-k+1], X[t-k+2], ..., X[t]$ if the hardware uses a step-by-step forward propagation.

```
[1] Kim, Sangyeob, et al. "C-DNN: A 24.5-85.8 TOPS/W complementary-deep-neural-network processor with heterogeneous CNN/SNN core architecture and forward-gradient-based sparsity generation." 2023 IEEE International Solid-State Circuits Conference (ISSCC). IEEE, 2023.
[2] Chang, Muya, et al. "A 73.53 TOPS/W 14.74 TOPS heterogeneous RRAM In-memory and SRAM near-memory SoC for hybrid frame and event-based target tracking." 2023 IEEE International Solid-State Circuits Conference (ISSCC). IEEE, 2023.
```

## Trade-off

#### **Disadvantages**

The PSN family lacks the neuronal reset. The neuronal reset is important in neuronal dynamics, as Table 1 has shown that the LIF neuron without reset has lower performance on temporal tasks. We summarize the effect of neuronal reset as follows:

1. Avoid too-high firing rates
2. Clear (hard reset) or reduce (soft reset) the influence of previous inputs after firing a spike
3. Introduce a nonlinearity during generating hidden states

For effect 1, we have shown in Figure 4 and lines 265-269 that the firing rate of the PSN family is higher than vanilla spiking neurons, but the increment degree is minor. And the firing rate is still far away from 1.0, which does not damage accuracy. Meanwhile, the learnable weights and thresholds of the PSN family can also adjust the firing rates directly.

Effect 2 works similarly to the gate mechanism in LSTMs. For the moment, the PSN family does not involve such a dynamic mechanism. The PSN simply uses all inputs without filtering, while the masked PSN and sliding PSN uses the latest $k$ inputs.

Effect 3 is caused by involving $S[t]$ in neuronal reset, which is generated by the nonlinear Heaviside function. While the generation of $H[t]$ in the PSN family is fully linear.



Although the PSN family lacks effects 2 and 3, it still works better in some temporal/static data classification tasks, as we have shown in this paper.
The PSN family is the prototype of parallelizable spiking neurons. Based on it, effects 2 and 3 can be implemented by introducing a nonlinear gate mechanism. Here let us provide the formulation of the gated PSN. For example, when adding the input gate $\mathbf{I}$ and the forgetting gate $\mathbf{G}$ as
$$
\mathbf{I} = \sigma(\mathbf{W_{I}}\mathbf{X} + \mathbf{B_{I}}), ~~~~~~~~~~~~~~~\mathbf{W_{I}} \in \mathbb{R}^{T \times T}, \mathbf{X} \in \mathbb{R}^{T \times N}, \mathbf{B_{I}} \in \mathbb{R}^{T}
$$

$$
	\mathbf{G} = \sigma(\mathbf{W_{G}}\mathbf{X} + \mathbf{B_{G}}), ~~~~~~~~~~~~~~~\mathbf{W_{G}} \in \mathbb{R}^{T \times T}, \mathbf{X} \in \mathbb{R}^{T \times N}, \mathbf{B_{G}} \in \mathbb{R}^{T}
$$

where $\sigma$ is the sigmoid function or the Heaviside function with surrogate gradients. Then the hidden states are generated by
$$
H[t] = G[t] \cdot H[t-1] + (1 - G[t])\cdot I[t] \cdot X[t].
$$
Although $H[t]$ is calculated by an iterative equation, it can still be parallelized by Parallel Prefix Sum (Scan) algorithm as
$$
p[i] = \prod_{j=0}^{i}G[j],
$$
$$
	c[i][j] = \prod_{l=i+1}^{j},
$$
$$
G[l] = \begin{cases}
			\frac{p[i]}{p[j]}, j \geq i \\\\
			0, \mathrm{otherwise}
		\end{cases},\\
$$

$$
	H[t] = \sum_{i=0}^{t}(1 - G[t])\cdot I[t] \cdot X[t] \cdot c[i][t].
$$

With CUDA devices to solve $H[t]$ at all $t$, the time complexity is $\mathcal{O}(\mathrm{log}(T))$.

#### **Advantages**

The advantages of the PSN family have been discussed in the paper in detail. We conclude them as:

1. Highly parallelizable neuronal dynamics and fast simulation speed
2. Higher or equal task performance to the vanilla spiking neurons
3. Easy to learn long-term dependency because the connection between any $X[i]$ to $H[j]$ is direct

## Network Structure Symbols

To answer some questions, we need to show the detailed network structure. We use the following symbols to represent the network structure in our responses:

- `c2k3s4`: the convolutional layer with output channels `2`, kernel size `3`, and stride `4`
- `BN`: the batch normalize layer
- `APk2s2`: the average pooling layer with kernel size `2` and stride `2`
- `Flatten`: the flatten layer
- `DP`: the dropout layer
- `FC10` as the fully connected layer with `10` output features
- `{}*2`: `2` repeat structures, e.g., `{FC10}*2` is `{FC10-FC10}`

---

> ### Author Response · Authors · 2023-08-13
> **Fix math errors**
>
> An additional line break causes the math errors of $c[i][j]$. It should be corrected as
>
>
> $$
> 	c[i][j] = \prod_{l=i+1}^{j}G[l] = \begin{cases}
> 			\frac{p[i]}{p[j]}, j \geq i \\\\
> 			0, \mathrm{otherwise}
> 		\end{cases},\\
> $$

---

### Decision · Program_Chairs · 2023-09-21

**Decision:**

Accept (poster)

**Comment:**

The reviewers were in agreement on the valuable contributions of this paper, and I commend the authors on both the paper and the detailed rebuttal, which was critical in winning over all reviewers.  I'm pleased to report that it has been accepted to NeurIPS.  Congratulations!  Please revise the manuscript according to the reviewer comments and discussion points.